# The Detection and Partial Localisation of Heteroplasmic Mutations in the Mitochondrial Genome of Patients with Diabetic Retinopathy

**DOI:** 10.3390/ijms20246259

**Published:** 2019-12-11

**Authors:** Afshan N. Malik, Hannah S. Rosa, Eliane S. de Menezes, Priyanka Tamang, Zaidi Hamid, Anita Naik, Chandani Kiran Parsade, Sobha Sivaprasad

**Affiliations:** 1Department of Diabetes, School of Life Course Sciences, Faculty of Life Science and Medicine, King’s College London, London SE1 1UL, UK; 2NIHR Moorfields Biomedical Research Centre, Moorfields Eye Hospital, London EC1V 2PD, UK

**Keywords:** diabetic retinopathy, mitochondrial DNA, surveyor nuclease, heteroplasmic mutations

## Abstract

Diabetic retinopathy (DR) is a common complication of diabetes and a major cause of acquired blindness in adults. Mitochondria are cellular organelles involved in energy production which contain mitochondrial DNA (mtDNA). We previously showed that levels of circulating mtDNA were dysregulated in DR patients, and there was some evidence of mtDNA damage. In the current project, our aim was to confirm the presence of, and determine the location and prevalence of, mtDNA mutation in DR. DNA isolated from peripheral blood from diabetes patients (*n* = 59) with and without DR was used to amplify specific mtDNA regions which were digested with surveyor nuclease S1 to determine the presence and location of heteroplasmic mtDNA mutations were present. An initial screen of the entire mtDNA genome of 6 DR patients detected a higher prevalence of mutations in amplicon P, covering nucleotides 14,443 to 1066 and spanning the control region. Further analysis of 42 subjects showed the presence of putative mutations in amplicon P in 36% (14/39) of DR subjects and in 10% (2/20) non-DR subjects. The prevalence of mutations in DR was not related to the severity of the disease. The detection of a high-prevalence of putative mtDNA mutations within a specific region of the mitochondrial genome supports the view that mtDNA damage contributes to DR. The exact location and functional impact of these mutations remains to be determined.

## 1. Introduction

Diabetic retinopathy (DR), one of the most common complications of diabetes, is the leading cause of visual impairment and blindness in the working age population [1]. Despite current prevention regimes targeting blood glucose control, blood pressure, and lipids, the incidence of DR remains high [2]. By the end of one decade of diagnosed diabetes, ~50% of type 1 diabetes and >30% of type 2 diabetes patients develop DR [3]. After two decades, a staggering 90% of those with type 1 and 60% of those with type 2 are likely to suffer visual impairment caused by DR [4]. With the epidemic rise in diabetes, affecting a reported 463 million people worldwide in 2019 and predicted to rise to 700 million by 2045, at least a third of the millions of people affected by diabetes will also develop DR making this a global and extremely urgent problem [5]. Therefore, there is a strong clinical, societal, and economic need to understand the molecular mechanisms involved and to design novel preventative treatment strategies for the development or progression of DR.

DR is asymptomatic in the initial years of diabetes, during which time molecular changes, currently undetectable in humans, precede visible microvascular changes. Such changes propagate with increased duration of diabetes, however, currently it is not possible to detect them before retinal damage. Screening programs require high-cost imaging tools to assess sight-threatening DR [2] but are unable to detect the very early stages of disease. Development of new therapeutic and preventive strategies are dependent on a better understanding of the pathways leading to disease, and the search for biomarkers to monitor disease progression and treatment responses has become crucial with the rising epidemic of diabetes.

Hyperglycaemia is a key contributor of damage to the eye, leading to increased intracellular reactive oxygen species (ROS), oxidative stress, and damage to DNA, proteins, and lipids [6]. Mitochondria, the site of energy production in the cell, are implicated in the overproduction of retinal ROS. Emerging evidence from animal and cell models in the past decade has highlighted mitochondrial dysfunction as a key player in the development of DR [7]. Mitochondria contain their own extranuclear DNA genome known as mitochondrial DNA (mtDNA), a 16.5 kb circular double-stranded DNA molecule which is present as multiple copies per cell throughout the body. Due to its proximity to the mitochondrial electron transport chain, mtDNA can be damaged more easily than nuclear DNA in conditions of oxidative stress.

In experimental models of DR (cultured cells and rodents), it has been shown that mtDNA damage precedes retinal damage [7,8,9] however, there have been few studies of these mechanisms in humans [10]. This is mainly due to the challenges of studying the damage to retinal mitochondria in humans due to the inaccessibility of suitable samples at progressive stages of disease. We previously proposed that blood samples from DR patients could be useful for the detection of systemic mitochondrial dysfunction. In a cross-sectional study comprising of 220 patients, we showed that circulating mtDNA levels were independently associated with DR, showing a biphasic response to the disease, and in parallel, we detected increased inflammation and some evidence of mtDNA damage in DR patients [11].

We previously proposed the hypothesis that early changes in mtDNA levels reflect subsequent mitochondrial dysfunction and mtDNA damage [12]. We postulated that in DR, hyperglycaemia initially causes a maladaptive response in the form of increased cellular mtDNA levels, and subsequently, a bioenergetic deficit through mtDNA damage and impaired oxidative phosphorylation (OXPHOS). We were able to detect increased mtDNA damage in 220 DR patients using a PCR-based method [11], however, although this was an indication of mtDNA mutations, there were limitations with the method. Blood cells can contain hundreds of copies of mtDNA per nuclear genome which, if damaged, can coexist with wildtype DNA in a situation known as heteroplasmy. Due to the coexistence of both mutant and wildtype mtDNA within the same cell, the methodology for the detection of mtDNA mutations is not straightforward and studies often use PCR-based methods which can give an indication of damage but do not provide information on the location. Although PCR-based methods have been used to detect mutations in mtDNA by us [11] and others [13], these methods can have a high degree of error and also are unable to distinguish between the types of DNA lesion. Therefore, in the current study, we used the surveyor nuclease method described by Bannwarth et al. [14]. This method uses the DNA repair enzyme Surveyor nuclease which can detect mismatch mutations in DNA samples with a heteroplasmy level as low as <3% in comparison to other mutation detection methods, such as DNA sequencing that detects mutations with a heteroplasmy level of <15% [15]. In effect, this means that low levels of mtDNA damage can be found and located to specific parts of the mitochondrial genome.

In the current study, we wanted to confirm and extend our earlier finding of mtDNA damage in DR patients [11]. Our aims were to (a) confirm the presence of these mutations in a larger cohort of patients, (b) determine if any mutations we find are randomly distributed throughout the mtDNA or whether there are hotspots, and (c) determine if mtDNA mutations in DR patients associate with severity of DR.

## 2. Results

### 2.1. Characteristics of Patients Used in This Study

This cross-sectional study was carried out using blood samples from 59 individuals (Table 1). Specifically, we compared two groups, 20 diabetes patients with no retinopathy at the time of this study (DR-0) and 39 diabetes patients with retinopathy (DR) at the time of this study. The DR patients had higher glycated haemoglobin (HBA1c) and higher estimated glomerular filtration rate (eGFR) but did not differ in terms of age, diabetes duration, body mass index (BMI), and other parameters. We had more type 2 diabetes patients than type 1 in the cohort, and whilst the gender ratio of females to males was higher in the DR-0 group (no retinopathy), the DR group had more males than females although this was not statistically significant.

### 2.2. Detection of Damage in 6 Overlapping Regions of the Mitochondrial Genome: Pilot Study

In order to determine potential hotspots of mtDNA damage, we first carried out a pilot study using DNA from 6 DR patients. Total DNA isolated from whole peripheral blood was used to amplify 6 regions of overlapping regions of mtDNA (regions designated as I, J, K, M, O, P), shown in Figure 1. The primers representing the 6 regions of mtDNA and their respective amplicon sizes were originally described by Bannwarth et al. [14] and are given in Table A1. For each region, the amplicons were generated by PCR, digested with Surveyor Nuclease, and electrophoresed on agarose gels against nondigested bands. The presence of putative mutations was identified by the detection of digestion products. Mutations were primarily focused and repeatedly detected in amplicon P and its overlapping amplicon region, amplicon H’. No mutations were detected in any of the other amplicons. In each case, putative mutations were confirmed by different individuals carrying out independent experiments, and by the identification of mutations in the overlapping amplicons. From these pilot experiments, we decided to focus on amplicon P of the mitochondrial genome for a larger study.

### 2.3. Prevalence of Mitochondrial DNA Mutations in Amplicon P in the Study Cohort

DNA samples from the entire cohort of 59 patients were used to amplify amplicon P and the overlapping amplicon H’. Mutations in amplicon P were only called if they could be replicated in independent experiments and also confirmed in the overlapping amplicon H’ giving the expected sized bands. An example of the detection of mutations in 3 patients is shown in Figure 2.

Although we had predicted mtDNA damage would occur randomly, we noted potential mutational hotspots, i.e., cleaved products of similar sizes in different patients, which we identified by matching bands across patients in relation to ladder and location of undigested band. Sixteen out of the 59 patients showed the presence of mtDNA mutations in amplicon P (Table 2). Analysis showed that blood samples from 10% of the DR-0 and 36% of the DR subjects contained putative mutations (Figure 3). There was no difference in the prevalence of suspected mutations in association with the severity of DR at the time of sample collection.

### 2.4. Does the Prevalence of Mutations Change with Severity of DR?

The DR group was subdivided as having mild DR (DR-m) or severe DR (DR-s), as we have previously described [11]. The two groups showed differences in mtDNA levels, the mtDNA content of the subjects with DR-m was higher than those with DR-s, suggesting dysregulation of mtDNA levels, as we have previously reported (Table 3). The DR-s group also had reduced HBA1c and eGFR relative to the DR-m group and mostly comprised of T2D patients. However, in terms of mtDNA damage, there was no difference in the prevalence of putative mutations between these two groups.

## 3. Discussion

We have previously shown that circulating mtDNA content was dysregulated in patients with DR, showing a biphasic change. Additionally, we reported putative mtDNA damage in 8 DR patients using the “elongase” method [13], which compares the efficiency of the amplification of a large fragment of mtDNA [11]. Based on this and other studies, we proposed a hypothesis in which an early maladaptive increase in mtDNA is involved in subsequent diabetic complications [16]. However, our previous data was only suggestive of damage and did not give any information about the prevalence or location of mtDNA damage in DR. In the current paper, our aim was to confirm and extend our earlier finding of mtDNA damage in patients with DR. In order to demonstrate mtDNA damage, the issue of heteroplasmy needs consideration, since mtDNA is present in cells as a multicopy molecule, ranging from hundreds to thousands of copies per cell. We hypothesized that blood cells from patients with DR may contain a mixture of mostly wildtype and a small percentage of mutated forms of mtDNA, the latter proposed to arise from hyperglycaemia-induced damage to mtDNA, which we postulated would be random and could affect any region of the 16.5 kb mtDNA genome.

Interestingly, a vast majority of studies evaluating mtDNA damage in diabetic complications have identified mtDNA damage by using the amplification of a long-fragment relative to a short-fragment in the mitochondrial DNA [16,17,18]. The principle of the amplification-based technique, originally described by Furda et al. [13], is the ability of DNA lesions to block the progression of the polymerase. In this regard, this technique does not allow for the identification of potential sequence-specific lesions or sequences that might not stall or stop the polymerase, i.e., lesions occurring within primer-annealing regions. Any method for detection needs to have the sensitivity to detect mutations present at low levels in a mixture containing both wildtype and mutant mtDNA molecules. For this reason, in the current study, we chose to use the surveyor nuclease method which can detect low levels of heteroplasmy and can also be used to broadly identify the region of mtDNA that may be damaged [14,15]. This mutation detection technique relies on the amplification of specific regions of the mtDNA genome as overlapping amplicons which are then digested to identify putative mismatch sites. For wildtype mtDNA, none of the amplicons should contain mismatch sites, whereas if mutations are present, then the enzyme will cleave at the mismatch site and the resulting digested products can be visualized as cleaved products. Any putative mismatch sites can be confirmed by the amplification of an overlapping amplicon, allowing for confirmation of the mutation. The sensitivity of this method has been reported to be 90%, which cleaves at sites of base-substitution mismatch and insertion/deletion of up to at least 12 nucleotides [19]. Screening of the entire mtDNA genome as 13 overlapping fragments, covering the mtDNA genome in a subset of 6 patients, showed that putative mutations were detected more often in amplicon P of the mitochondrial genome. Amplicon P comprises a number of noncoding regions for tRNAs, and the genes encoding ND6, cytochrome B, and part of 12S ribosomal RNA (Figure 4). When we undertook a more detailed analysis of amplicon P, we found that 10% of the DR-0 patients and 36% of the DR patients examined had heteroplasmic mutations in one or more locations in amplicon P. Surprisingly, many patients had very similar cleavage patterns, suggesting mtDNA mutations which may be hotspots.

Amplicon P contains a number of genes encoding protein subunits of the electron transport chain, tRNAs and an rRNA molecule. The NADH dehydrogenase 6 (MT-ND6) gene encodes a subunit of complex I, a significant part of the electron transport chain in the inner mitochondrial membrane [20]. Mutations in MT-ND6 have been found in association with several diseases, including Leber’s Hereditary Optic Neuropathy (LHON), where retinal ganglion cells are affected leading to loss of visual acuity, and Leigh Syndrome, a primarily neurological condition with paediatric onset [21]. Cytochrome B (MT-CYB) is one of 11 proteins forming complex III of the electron transport chain, and mutations have been linked to muscle weakness [22]. Mutations in 12S rRNA (MT-RNR1), a component of the small subunit of the mitochondrial ribosome, have been associated with sensorineural hearing loss [23]. Several pathogenic mutations have been identified in genes encoding tRNAs located within the region encompassed by amplicon P, namely, tRNA phenylalanine [24], proline [25], threonine [26], and glutamic acid [27]. The control region is a hypervariable region of noncoding DNA which has been used to trace human population lineages [28]. Mutations in this region have been identified in several cancers [29,30,31].

Diabetic retinopathy, a common complication of diabetes, is not generally regarded as a mitochondrial disease. Traditionally, mitochondrial genetic diseases arise due to mutations in mtDNA itself or in nuclear genes required for mitochondrial function. These diseases are rare, affecting approximately 1 in 4300 adults, and can affect multiple cell types and organ systems leading to symptoms with a range of severity and onset at any age [32]. However, an increasing body of evidence suggests a link between mitochondrial dysfunction and diabetic complications [16,33], which, due to the epidemic rise in diabetes, are prevalent affecting millions of people worldwide. Chronic exposure of human retinal cells to hyperglycaemia has been shown to induce mtDNA oxidative damage and cell death *in-vivo* and *in-vitro* [34,35,36].

The possibility that mtDNA damage is involved in diabetic complications has been proposed by us previously in work that lead directly to the current study [11,12,16], where we showed using *in-vitro* studies that there is an early maladaptive increase in mtDNA which precedes mtDNA damage and bioenergetic deficit [12]. A similar observation was made in gestational diabetes where an increase in mtDNA copy number was seen in parallel with an increase in mtDNA oxidative lesions in gestational diabetes cases versus control pregnancies [18]. Similarly, in diabetic kidney disease, increased evidence of oxidative mtDNA lesions has been highlighted, where mtDNA has been found to be increasingly damaged in renal biopsies, urine, and in serum [17,37]. Our prior studies in diabetic nephropathy also demonstrate this association between mitochondrial dysfunction and diabetic complications, where we saw potential evidence of mtDNA damage in diabetic nephropathy patients versus healthy controls, though this was not confirmed by a secondary technique [12].

In relation to DR, mtDNA damage was detected using the elongase method in a relatively small cross-sectional study involving 11 nondiabetic healthy controls and 11 diabetic individuals, of which 6 were diagnosed with DR, 3 of which had additional microvascular complications such as neuropathy and nephropathy [35]. Interestingly, no mutations were seen in diabetes patients without DR in this study, whereas we report potential mutations in 10% of our DR-0 cohort. A small cohort study of 10 individuals with late diabetic complications including retinopathy, nephropathy, neuropathy, and other macrovascular complications were found to have increased evidence of mtDNA lesions in peripheral blood mononuclear cells in comparison to 10 age-matched healthy volunteers [38]. Whilst we did not evaluate healthy controls as part of our current study, the literature supports our finding of increased mtDNA damage in diabetes and its complications.

The surveyor nuclease method was used in this study for qualitative analyses of mtDNA mutations, i.e., we can report on the presence/absence of potential mutations, but we cannot determine the precise locations of these mutations or the level of heteroplasmy at which they may be present. Next generation sequencing could be useful for verifying our findings and this will enable us to better classify the nature of these mutations. Furthermore, whilst this study features a larger cohort size than many others in the literature, we have not been able to stringently evaluate the relationship between mtDNA damage and diabetes type, the presence of additional microvascular complications, or the clinical progression of DR. In addition, healthy controls without diabetes were not available at the time of this study.

In summary, we show evidence of increased prevalence of mtDNA damage in circulation in DR patients. We have identified potential mtDNA mutations in a region spanning genes encoding ND6, cytochrome B, 12S ribosomal RNA, a number of tRNAs, and the noncoding region of the mitochondrial genome. The functional impact of such mutations is not yet known. Mutations were more common in those with DR versus those without. Therefore, our data suggests that mtDNA mutation is associated with DR, however, the nature of this association is yet to be fully elucidated. Damaged out of place mtDNA may directly contribute to increased inflammation in DR-s as it resembles bacterial DNA. If damaged mtDNA contributes to the progression of retinopathy, then the design of strategies for the removal of damaged circulating mtDNA could provide a novel therapeutic target for the treatment and prevention of diabetic retinopathy progression.

## 4. Materials and Methods

### 4.1. Subjects

Patients were recruited from King’s College Hospital (NHS Research Ethics Committee approval REC; ref number08/H0808/228) and Guy’s and St Thomas’s hospital clinics (regional Research Ethics Committee REC ref number 07/H0806/120) (Project code was KCH1039, approval date 07 May 2009) with written informed consent in accordance with the Declaration of Helsinki. Categories for DR severity were defined using the Early Treatment Diabetic Retinopathy Study (ETDRS) [31] severity system as previously defined [11].

Metabolic status (HbA1c) and lipid levels were assessed by the hospital clinical lab services using standardized assays. The Jaffe assay was used to measure creatinine. Glomerular Filtration Rate (GFR) was assessed using the Modification of Diet in Renal Disease (MDRD) formula [39]. Body mass index (BMI, kg/m^2^), diastolic blood pressure, and systolic blood pressure (mmHg) were recorded during the visit to the clinic at the time of blood sampling.

### 4.2. Determination of Circulating mtDNA Content

Whole blood/buffy coat was obtained and frozen within 2 h of collection at −20 °C and transferred to −80 °C within 24 h. Total genomic DNA was isolated from 100 µL thawed sample using the DNeasy Blood and Tissue Kit (Qiagen, Manchester, UK) and treated with sonication as described in [40]. MtDNA content was determined using real time quantitative PCR as previously described [11].

### 4.3. Detection of mtDNA Damage Using the Surveyor Nuclease Method

Potential mutations in mtDNA were detected using the surveyor nuclease method [14]. First, primer pairs were used to amplify overlapping amplicons as follows: 1 µL of template DNA was added to a mastermix containing 2× DreamTaq mastermix (ThermoFisher, Paisley, UK), forward and reverse primers (200 nM each) made up to a final volume of 50 µL with ddH2O. Thermocycler conditions for this reaction were as follows: 94 °C for 15 min; 30 cycles of 94 °C for 30 s, 60 °C for 30 s, and 72 °C for 90 s; 72 °C for 7 min; hold at 4 °C. Amplicons were checked by agarose gel electrophoresis and stored at −20 °C until future use.

Nuclease digestion of these amplicons was carried out using the Surveyor Nuclease Mutation Detection Kit (Integrated DNA Technologies, Belgium). A mastermix containing Optimase Polymerase Buffer (6 µL), MgCl2 (1.5 µL), Surveyor enhancer (1 µL), Surveyor Nuclease (1 µL), and ddH2O (35.5 µL) was prepared on ice. For each amplicon, ‘digested’ and ‘undigested’ reactions were prepared by adding 15-µL PCR product to 45-µL mastermix (digested) or 51-µL ddH2O (undigested). Digested reaction mixtures were incubated at 42 °C for 35 min while undigested samples were left at 20 °C. A total of 6 µL of stop solution was added to the digestion mixture to stop the nuclease activity. Products were visualised on 1.5% agarose gel containing ethidium bromide at 80 mV for 40 min.

### 4.4. Statistical Analyses

Statistical analyses of cohort characteristics were performed using GraphPad prism (v8.0.0 for Windows, GraphPad Software, San Diego, Ca, USA) and IBM SPSS (v25.0.0.1 for windows, Armonk, NY, USA). Nonparametric Mann–Whitney *U* test was used to compare DR groups after dataset normality was assessed by Shapiro–Wilk’s *W* test (*p* < 0.05, suggesting non-normality).

## 5. Conclusions

We have shown increased prevalence of mutations in a specific region of the mitochondrial genome in blood samples from patients with DR, and a lower prevalence in diabetes patients without DR. Our data suggests that there may be hotspots of mtDNA damage. The exact nature of these mutations needs to be established in order to determine their potential functional impact. Our data add to the growing body of evidence suggesting that mtDNA is affected in diabetic complications and that acquired hyperglycaemia-induced mtDNA damage may be involved in disease progression. Longitudinal studies could be undertaken to determine if mtDNA is damaged as a consequence of diabetes and whether specific regions of damage contribute to the pathogenesis of DR.

## Figures and Tables

**Figure 1 ijms-20-06259-f001:**
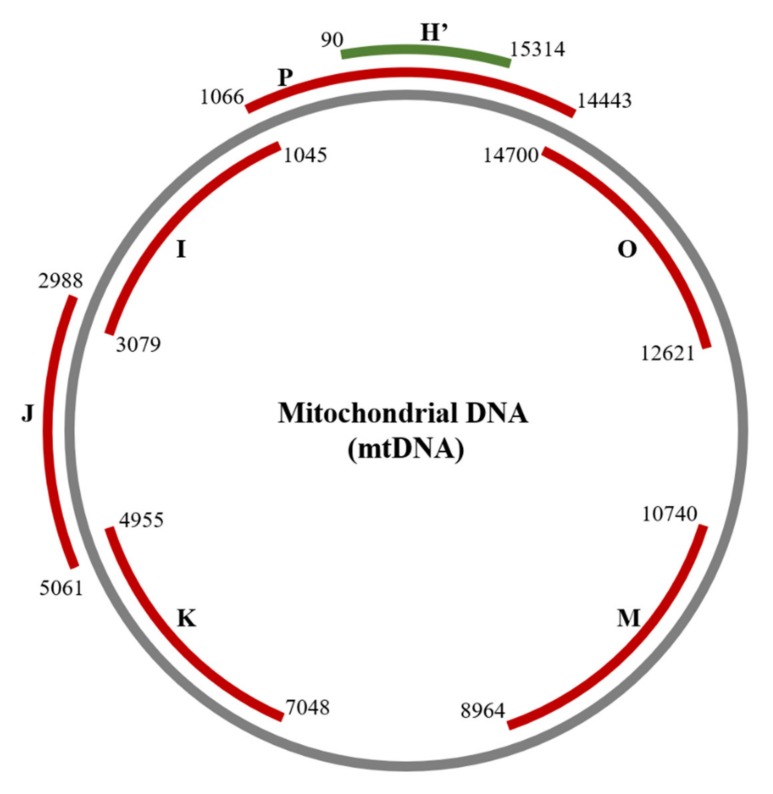
Overlapping amplicons of the mitochondrial DNA genome. Amplicons P, I, J, K, M, and O (red lines) were generated from total genomic DNA isolated from whole blood of diabetes patient with and without retinopathy. A pilot experiment highlighted amplicon P as a site of suspected mtDNA mutations, which were investigated by amplification of the overlapping amplicon H’ (green line). Primers used were described by Bannwarth et al. [14] (see Table A1). Nucleotide locations marked correspond to GenBank J01415.1.

**Figure 2 ijms-20-06259-f002:**
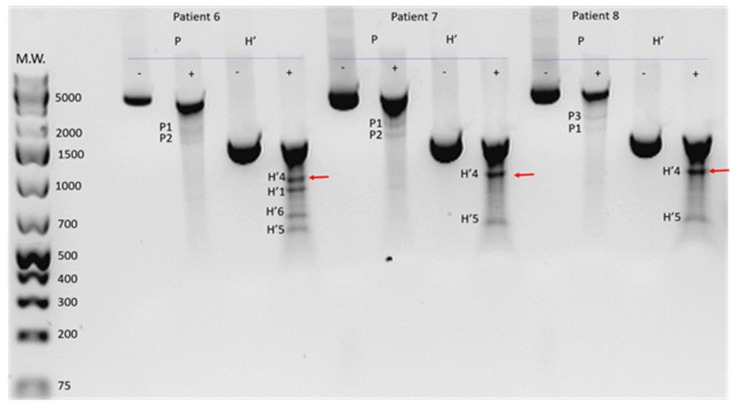
Detection of putative mitochondrial DNA mutations using the Surveyor Nuclease method. Total genomic DNA was isolated from whole blood and underwent PCR with primers for amplicon P and overlapping amplicon H’. Agarose gel electrophoresis of undigested (-) and digested (+) amplicons P and H’ from patients 6,7,8 show the presence of multiple bands, indicating a potential mutation is present. Bands were labelled P1–P3 and H’1–H’7, according to their relative amplicons, to identify products of the same size in different samples, i.e., fragment H’4 (red arrow) is present in amplicon H’ of patient 6, 7, and 8. M.W: molecular weight marker GeneRuler 1 kb plus (ThermoFisher).

**Figure 3 ijms-20-06259-f003:**
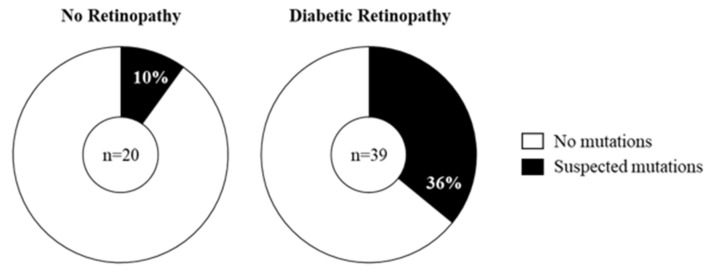
Prevalence of suspected mtDNA mutations in the study cohort. MtDNA mutations were identified by the amplification of overlapping regions (amplicons P and H’) of the mtDNA genome followed by digestion of mismatch-specific DNA endonuclease Surveyor Nuclease. Cleaved products were separated and analysed by gel electrophoresis with mutations identified by the presence of digested products. Suspected mutations were present in 2 (10%) diabetes patients without retinopathy (DR-0, *n* = 20) and 14 (36%) patients with diabetic retinopathy (DR-m and DR-s combined, *n* = 39) as shown by the black shaded region.

**Figure 4 ijms-20-06259-f004:**
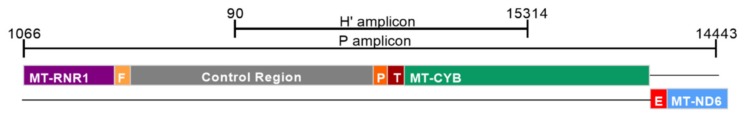
The region of the mitochondrial genome representing amplicon P. A schematic showing the genes present within amplicon P and the overlapping amplicon H’ where mutations were detected in blood from DR patients. MT-RNR1 (encodes 12S rRNA), F (tRNA phenylalanine), control region (noncoding), P (tRNA proline), T (tRNA threonine), MT-CYTB (cytochrome B) are located on the heavy strand, E (tRNA glutamic acid) and MT-ND6 (NADH dehydrogenase subunit 6) are located on the light strand. Nucleotide positions are given for amplicons P and H’ on the mitochondrial genome (NC_012920.1).

**Table 1 ijms-20-06259-t001:** Patient characteristics.

	Diabetes with No Retinopathy (DR-0)	Diabetic Retinopathy (DR)	*p*-Value
Number of patients	20	39	
Age (years)	66.1 ± 15.9 (31.0–85.0)	58.8 ± 14.0 (30.0–82.0)	ns
Sex (male:female)	7:13	22:17	ns
Diabetes (type 1:type 2)	5:15	9:30	
Duration (years)	13.0 ± 8.6 (5–43)	17.7 ± 10.8 (1–50)	ns
BMI (kg/m^2^)	28.7 ± 4.1 (21.9–34.7)	31.0 ± 12.1 (20.0–82.7)	ns ^a^
HbA1c (%)	7.5 ± 1.2 (5.7–9.5)	8.8 ± 2.1 (5.8–13.1)	*p* = 0.0127
HbA1c (mmol/mol)	58.2 ± 13.4 (38.8–80.3)	72.6 ± 22.5 (39.9–119.7)	*p* = 0.0126
eGFR (mL·min^−1^·1.73 m^−2^)	61.1 ± 22.7 (17.0–98.0)	77.3 ± 26.7 (10.0–161.0)	*p* = 0.0331 ^a^
ACR (mg/mmol)	2.9 ± 5.0 (0.1–20.8)	7.2 ± 14.3 (0.2–51.9)	ns ^a^
Systolic BP (mmHg)	133.9 ± 19.0 (125.0–187.0)	132.4 ± 16.3 (97.0–169.0)	ns
Diastolic BP (mmHg)	75.5 ± 8.1 (60.0–97.0)	74.8 ± 9.2 (56.0–96.0)	ns
Total cholesterol (mmol/L)	4.3 ± 0.8 (2.9–5.8)	4.3 ± 1.0 (2.3–7.4)	ns
mtDNA content (Mt/N)	105.2 ± 54.5 (33.3–208.1)	131.9 ± 117.3 (20.9–624.7)	ns

BMI: body mass index; HbA1c: glycated haemoglobin; eGFR: estimated glomerular filtration rate; ACR; albumin creatinine ratio (urine); BP: blood pressure; mtDNA: mitochondrial DNA; Mt/N: mitochondrial DNA copies per cell. ns: nonsignificant. Values shown are mean ± SD (range). *p*-values derived from unpaired *t*-test on normally distributed data unless otherwise stated. ^a^ Mann Whitney *U* test (nonparametric).

**Table 2 ijms-20-06259-t002:** Overlapping mutations suggest possible mutational hotspots.

Category	Patient	P Amplicon					H’ Amplicon						
		P1	P2	P3	P4	P5	H’1	H’2	H’3	H’4	H’5	H’6	H’7
DR-0	114	+	+	-	-	-	+	-	+	+	+	-	-
	153	+	+	-	-	-	+	-	+	+	-	-	-
DR-m	137	+	+	-	-	-	+	+	-	-	-	-	-
	157	+	+	-	-	-	+	+	-	-	-	-	-
	151	+	-	+	-	-	-	-	-	+	+	-	-
	91	+	+	-	-	+	+	-	-	+	+	+	-
	136	+	+	-	+	-	-	-	-	+	+	-	-
	110	+	-	+	+	-	-	-	-	+	+	-	-
	58	-	+	+	-	-	-	-	-	+	-	-	-
DR-s	161	-	-	+	-	-	-	-	-	+	-	-	-
	65	+	+	-	-	-	-	-	-	+	-	-	-
	14	+	+	-	-	-	-	-	-	+	+	-	+
	16	+	+	-	-	-	-	-	-	-	+	+	+
	8	+	-	+	-	-	-	-	-	+	-	+	+
	9	+	-	+	-	-	+	-	-	+	-	-	-
	23	+	+	-	-	-	n	n	n	n	n	n	n

DR-0: Diabetes patients with no retinopathy, ETDRS < 20; DR-m: mild nonproliferative diabetic retinopathy, ETDRS 35-43; DR-s: severe nonproliferative and proliferative diabetic retinopathy, ETDRS ≥ 47; n: not measured. The presence of possible mutations detected in 2 overlapping amplicons is indicated by the grey box with +. Mutations not seen in overlapping amplicons are not shown.

**Table 3 ijms-20-06259-t003:** Diabetes patients with mild and severe retinopathy.

	Mild Diabetic Retinopathy (DR-m)	Severe Diabetic Retinopathy (DR-s)	*p* Value
N =	20	19	
Age (years)	54.7 ± 15.1 (30.0–82.0)	63.1 ± 11.5 (43.0–82.0)	ns
Sex (male:female)	10:10	12:7	ns
Diabetes (type1: type 2)	8:12	1:18	
Duration (years)	19.0 ± 12.4 (1–50)	16.4 ± 8.9 (1–33)	ns
BMI (kg/m^2^)	32.0 ± 10.4 (20.3–59.6)	30.0 ± 14.0 (20.0–82.7)	ns ^a^
HbA1c (%)	8.2 ± 2.2 (5.8–12.1)	9.4 ± 1.7 (6.3–13.1)	ns
HbA1c (mmol/mol)	66.5 ± 23.6 (39.9–108.7)	80.8 ± 18.6 (45.4–119.7)	ns
eGFR (mL·min^−1^·1.73 m^−2^)	86.6 ± 26.4 (29.0–161.0)	66.1 ± 22.9 (10.0–91.0)	*p* = 0.0297 ^a^
ACR (mg/mmol)	5.5 ± 13.2 (0.2–51.0)	9.9 ± 16.3 (0.5–51.9)	ns ^a^
Systolic BP (mmHg)	127.8 ± 15.6 (97.0–169.0)	137.5 ± 15.6 (108.0–169.0)	ns
Diastolic BP (mmHg)	74.4 ± 8.2 (56.0–89.0)	75.3 ± 40.5 (60.0–96.0)	ns
Total cholesterol (mmol/L)	4.3 ± 1.2 (2.3–7.4)	4.6 ± 1.3 (3.2–7.4)	ns
mtDNA content (Mt/N)	183.0 ± 139.8 (20.8–624.7)	78.2 ± 50.1 (33.1–183.0)	*p* = 0.0063 ^a^

BMI: body mass index; HbA1c: glycated haemoglobin; eGFR: estimated glomerular filtration rate; ACR; albumin creatinine ratio (urine); BP: blood pressure; mtDNA: mitochondrial DNA; Mt/N: mitochondrial DNA copies per cell. ns: nonsignificant. Values shown are mean ± SD (range). *p*-values derived from unpaired *t*-test on untransformed, normally distributed data unless otherwise stated. ^a^ Mann Whitney *U* test (nonparametric).

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
