# Peer review of "The Detection and Partial Localisation of Heteroplasmic Mutations in the Mitochondrial Genome of Patients with Diabetic Retinopathy"

_ijms, 2019, doi:10.3390/ijms20246259_

Round 1

Reviewer 1 Report

Malik and others have previously published a number of papers building up a mechanistic story of what happens in Diabetic Retinopathy. Here Malik et al use a large cohort of patients and well designed experiments to demonstrate that mutations in diabetic retinopathy tend to occur most commonly in a region of the genome spanning 14,443 to 1066. This is an important finding and useful addition to their previous work as well as important for ongoing work to understand the consequences of this. This is particularly so since, as they point out, much of the previous work has not been done in human samples. However, I think some changes to the way in which this work is described and presented need to be made prior to publication.

On page 2 of the introduction line 25 the authors state their hypothesis. The authors then state that to test this hypothesis they plan to test if there is an increase in mtDNA mutations in DR patients. However, this experiment does not test this hypothesis, in fact their previous work referenced in the discussion (Ref36) tests this hypothesis. The current work is a continuation of this investigation where they look to see if the mtDNA mutations are randomly distributed throughout the mtDNA or whether there are hotspots. Please adjust the introduction to introduce the prior work and adjust the aims to accurately reflect this.

Reviewer 2 Report

 The study by Malik et al. presents the deletion and partial localization of heteroblastic mutations in the mitochondrial genome of patients with diabetic retinopathy. The study is exciting and offers an important observation that could open new avenues for novel treatment strategies in DR. 

The authors have compared the mutations in mtDNA in diabetic patients with no retinopathy, and patients with diabetic retinopathy. Why is no age-matched healthy control included in the study? It is important to delineate that the mutations observed here are specific to DR, and will increase the significance of the study.  
